# Assessment of the Influence of Selected Technological Parameters on the Morphology Parameters of the Cutting Surfaces of the Hardox 500 Material Cut by Abrasive Water Jet Technology

**DOI:** 10.3390/ma15041381

**Published:** 2022-02-13

**Authors:** Tibor Krenicky, Stefania Olejarova, Milos Servatka

**Affiliations:** 1Department of Technical Systems Design and Monitoring, Faculty of Manufacturing Technologies with a Seat in Presov, Technical University of Kosice, Sturova 31, 08001 Presov, Slovakia; stefania.olejrova@tuke.sk; 2IMSLOV, P. Horova 19, 08001 Presov, Slovakia; ms@imslov.sk

**Keywords:** abrasive water jet, material cutting, surface quality, cut profile, regression model, Hardox, technological head feed rate, mass flow of abrasive

## Abstract

This article deals with the evaluation of selected aspects of abrasive water jet technology (AWJ) in the cutting of abrasion-resistant steel (HARDOX 500) with a thickness of 40 mm. The high abrasion resistance as a typical significant property of this steel ranks it among the special materials that are increasingly used. As the AWJ is a multiparametric technology, selected levels of feed rate, abrasive mass flow and pump working pressure were used in the experiments from the spectrum of technological parameters. For the purposes of evaluation, the examined cut surfaces were documented by a modified photographic method of displaying the cut surface by means of side lighting on the untreated cutting surface. The experimental part evaluates the dependences of selected cutting surface quality parameters (surface roughness and abrasive water jet deflection) on selected important technological parameters of the production system with AWJ technology (abrasive mass flow, technological head feed rate and pump working pressure). Based on the evaluation of the experiments, regression models were created to interpolate and extrapolate data to compare or supplement existing solutions in the field of research and as a basis for optimizing operating costs and increasing the efficiency of production systems with abrasive water jet technology.

## 1. Introduction

Abrasive water jet technology (AWJ) is a progressive tool used to machine a wide range of materials. Profiting from its several unique features, such as cold machining of hard-to-machine materials without their deformation, separation of inhomogeneous materials and a small amount of waste material, AWJ has a significant potential to replace other machining techniques in several applications [1]. The pioneer scientists dealing with the topic were Hashish [2] and Zeng and Kim [3]. Later, numerous investigations aimed at the machining process appeared, e.g., by Kovacevic and Yong [4,5]. The current state of research of abrasive water jet technology revealed that one of the most important problems is the quantification and modeling of the influence of technological parameters on surface quality parameters, particularly at hard-to-machine materials sheets by Savkovic [6]. Abrasive water-jet cutting is a multiparametric process where the quality of the output characteristics relies on the inputs. This has been proven by various experiments and theoretical analyses. Evaluation of cutting quantity and quality has been continuously studied by various groups [7,8,9,10]. Sutowska et al. [11] studied the influence of cutting parameters on kerf quality in detail. Some of the recent experiments were performed on Hardox^TM^ 400, 450 and 500 steel sheets by Filip et al. [12,13]. Yet there are numerous attempts at creating the models that characterize the cut area quality under specific conditions regarding material type and thickness, technological head feed rate, the abrasive mass flow rate, grain size and many others. Evaluation of the cutting quality is related to the quality of the cut walls. Although there is a constantly growing set of developed solutions to the problem, including methodologies and evaluations of experiments valid for specific measurement conditions, the current solutions still do not cover several variations.

Models of machining materials using AWJ technology describe the action of a concentrated water jet, the efficiency of which is often increased by doping with solid abrasive particles such as garnet [14]. The model proposed by Monno and Ravasio [15] is based on the assessment of the striation formation that depends mainly on the jet instability caused by vibrations during the cutting process. Similar to other high-energy beam technologies, the AWJ jet generates visible striations on the machined surface [16,17,18,19,20]. The quality of the cutting process is the result of the tool’s operation as part of the overall effect on the overall quality of the product, conditioned by three types of accuracy: shape, dimensional and surface measure, characterized mainly by roughness parameters [21]. The roughness of the machined surface using AWJ deteriorates from the point of entry into the material to the point of exit. The striating is generated at a certain depth below the surface and gradually deepens, causing a negative effect on the quality of the machined surfaces as well as on the shape accuracy of the products. The machined surface is thus divided into a smooth zone and a rough–striated zone. This behavior arises from the fact that since the jet penetrates the material, it gradually loses its kinetic energy and deflects. The relatively smooth area in the upper part of the cut is identified as the zone of cutting wear of the material, while the second striated area in the lower part of the cut arises as a consequence of deformation wear during cutting by AWJ technology [22,23].

The most common characteristics used for the evaluation of the surface roughness are *Ra*, the mean arithmetic deviation of the profile, and *Rz*, the height of the profile unevenness. These two quantities can be measured by contact profilometers or by non-contact profilometers [24,25,26]. Nevertheless, the values depend not only on cut material or depth in the kerf but also on abrasive material quality and grain size [27]. Hlaváč’s group had presented a different approach to the determination of the cutting wall quality than the use of the *Ra* and *Rz* values, proposing a direct relationship between the declination angle (measured between the tangent to the striation curve in the definite depth *h* and the impinging jet axis) and respective cutting wall quality [28,29]. Understanding the influence of machining conditions on the quality of the obtained cuts enhances the quality and effectiveness of cutting. The basic process parameters characterizing the machining of materials using AWJ include the pumping pressure; feed rate of the technological head; abrasive mass flow; diameter of the nozzle and focusing tube and distance of the technological head from the material surface [30]. The microscopic models describing the mechanism of material cutting were prepared as well as the macroscopic model of cutting front behavior [31,32].

The recent research is focused on complementing existing models and preparing some new ones that would be simple enough to be applicable in industrial conditions to help predict and control the production quality. The results presented in this paper can be used for the regression models helping predict the surface quality relationship with the cutting factors, such as water pressure, feed rate and abrasive mass flow rate. In our opinion, the study of parameters in presented combination and range are unique. Thus, the novelty of the present manuscript is based on a unique combination of the thick durable material and variated operational parameters. Our work is aimed to complement data for models present in some other research works as some readers maybe appreciate such kind of information. Authors hope that data can be useful from research as well as practical points of view.

## 2. Materials and Methods

The experimental cutting of test specimens was performed in the Liquid Jet Laboratory, Institute of Physics, Faculty of Mining and Geology, University of Mining—Technical University of Ostrava using a production system with AWJ technology. The system includes a Flow HSQ 5X high-pressure pump and an X-Y WJ1020-1Z-EKO workbench together with an X-Y CNC control system with a PaserIII^T^^M^ cutting head.

All samples were cut from Hardox^TM^ 500 abrasion-resistant plates with a nominal hardness of 500 HBW developed for applications with high demands on abrasion resistance. Material properties were obtained by a combination of quenching and tempering performed by manufacturer SSAB Oxelösund AB, Sweden. A sheet thickness of 40 mm was used for the individual sets of experiments.

The samples were cut from sheet metal at combinations of feed rate *v* (10; 15 and 20 mm/min), abrasive mass flow *m_A_* (170; 220 and 270 g/min) and pump working pressure *p* (300; 340 and 380 MPa). The plate was placed on the grid of the X-Y workbench, the mutually perpendicular sides of the plate being parallel to the working axes X and Y of the table. The distance of the nozzle of the cutting head from the material surface at all cuts was 2 mm. The process of cutting itself consisted of 2 parts—the production of the hole in the material from which the water jet began to cut, and consequently, cutting the samples in the form of an equilateral triangle (Figure 1a). All holes were made before cutting, while the cutting head held on its body a protective sheet metal element in the shape of a cylinder closed on the top that was designed to prevent spraying the reflected jet into the space and thus polluting the laboratory with droplets of water and abrasive (Figure 1b).

Cutting parameters were as following:

Water orifice diameter *d_o_*—0.25 mm

Stand-off distance *L*—2 mm

Focusing tube diameter *d_a_*—1.02 mm

Focusing tube length *l_a_*—76 mm

Abrasive material average grain size *a_o_*—0.275 mm (MESH 80)

Abrasive material—Australian garnet GMA

A total of 9 pieces of samples were cut. The total number of various cutting parameter combinations was 27 (3 areas on each sample). All the samples cut by abrasive water jet were marked for accurate evaluation to avoid confusion. The marking was performed immediately after cutting the set of samples and drying them on their upper surfaces (closer to the upper cutting edges). The beginning of the cut was marked with a dot on each upper surface of the sample. The arrow indicated the cutting direction and procedure. The method of labeling the samples for evaluation can be seen in Figure 2.

The roughness parameters *Ra* and *Rz* were measured in the middle height of the sample, i.e., at half the cut material’s thickness. The roughness parameters *Ra*_4_ and *Rz*_4_ were measured on the cut surfaces of samples at a distance of 4 mm from the upper cutting edge (from the surface of the sheet where the jet enters the material) using the Mitutoyo Surftest SJ-301 roughness tester—see Figure 3. Repeated control measurements were performed for the reliability of all measured sets of values. The control measurements’ total errors for the roughness *Ra*, *Rz*, *Ra*_4_, and *Rz*_4_ are in the range <3.06; 5.09> percent [15].

The deflection of the abrasive water jet *Ø* was measured at 5 locations on each cut surface, steadily at 5 mm distance from the previous measurement in the cutting direction according to Figure 4.

### 2.1. Description of Measuring Equipment

Roughness measurements of the cut surfaces of the samples were performed using the Mitutoyo Surftest SJ-301 roughness tester. The device was used to measure the surface roughness of the cut surfaces (*Ra*, *Rz*, *Ra*_4_ and *Rz*_4_) of the samples. The device contains a contact probe, which measures the surface profile using a differential induction detection method and evaluates the surface quality calculating parameters according to the standards. The length of the measuring needle path in the roughness measuring device was 12.5 mm. The detail of the measuring device when measuring the roughness of the cut surface of the sample is illustrated in Figure 5.

### 2.2. Analysis and Evaluation of the Surface

The analysis of the results of the experimental studies was structured into the following blocks:−Measured values of roughness *Ra*, *Rz*, *Ra*_4_ and *Rz*_4_ and deflection of the jet *Ø*;−Roughness measured in the middle of the cut areas of the samples;−Roughness measured 4 mm from the upper cutting edges on the samples’ cut surfaces;−Deflection of the abrasive water jet;−Evaluation based on photographic pictures of cut surfaces.

## 3. Results

### 3.1. Measured Values of Roughness Ra, Rz, Ra_4_ and Rz_4_ and Deflection of the Abrasive Water Jet Ø

Table 1 presents a set of measured values of the deflection of the abrasive water jet *Ø* and the surface roughness parameters *Ra* and *Rz* measured in the centers of the cut areas of the samples according to locations depicted in Figure 3 and Figure 4, varying with change in the selected technological parameters—feed rate of the technological head, abrasive mass flow and pumping pressure.

Table 2 presents a set of measured values of roughness parameters *Ra*_4_ and *Rz*_4_ of cut surfaces measured at a distance of 4 mm from the upper cutting edge according to Figure 3.

The values are varying with change in the selected technological parameters—feed rate of the technological head, abrasive mass flow and pumping pressure.

### 3.2. Roughness Measured in the Centers of the Cut Areas of the Samples

The graphs in Figure 6 present a graphical evaluation of 1-parametric dependencies of *Ra* = f(*v*). These dependencies are presented in the form of conjugated graphs for cases of constant abrasive mass flow (in each graph, three dependencies are plotted for the three different values of pressures of 300, 340 and 380 MPa).

The graphs in Figure 7 present a graphical evaluation of 1-parametric dependencies of *Rz* = f(*v*). These dependencies are shown in the form of conjugated graphs for cases of constant abrasive mass flow (in each graph, three dependencies are plotted for the three different values of pressures 300, 340 and 380 MPa).

### 3.3. Roughness Measured 4 mm from the Upper Cutting Edges of the Cut Areas of the Samples

Graphical evaluation of the 1-parametric dependence in Figure 8 shows the dependence of the technological parameter v on the quality parameters of the cut surface *Ra*_4_ at a constant distance of the roughness measuring point from the upper cutting edge as presented in Figure 3. The mass flow of the abrasive is constant. In each graph, three dependencies are plotted for the three different values of pressures 300, 340 and 380 MPa.

The graphs in Figure 9 show a graphical evaluation of the 1-parametric dependencies of *Rz*_4_ = f(*v*) at a constant distance of the roughness measuring point from the upper cutting edge (Figure 3). These dependencies are presented in the form of conjugated graphs for cases of the constant abrasive mass flow (in each graph, three dependencies are plotted for three different values of pressures of 300, 340 and 380 MPa).

### 3.4. Evaluation Based on Photographic Images of Cut Surfaces

The acquisition of a set of digital images from the cut areas of the examined samples was provided by a specific photographing method using side lighting to illuminate the untreated cut area, enabling jet deflection measurements. Figure 10 provides an example of four selected images of the cut surfaces taken by the above-mentioned method.

The observations made using the photographic pictures complement the previous information gained from experimental data evaluation and enable obtaining a more complete and clearer view of the specific technological parameters effect. Observing the lower parts of the cut surfaces of selected samples presented in Figure 10 (at the values of technological parameters *m_A_* and *p* from their lower ranges and *v* from the upper range), a markedly rough surface is visible, characterized by large bended striations caused by significant deflection (lagging) of the water jet and significant depth difference between the peaks and valleys of the striations.

All cut areas of the sample set show a significantly lower roughness in their upper part (closer to the cutting edge on the jet entering side) as compared with the lower part. Striations are visible on the cut surfaces of all samples. With more favorable values of technological parameters (lower *v*, higher *p* and higher *m_A_*) from the experimental set, it is possible to observe a gradual straightening of striations and simultaneously a reduction in the deflection of the abrasive water jet. The beginnings of striations are transferred with more favorable values of technological parameters from the upper part of the cut area to the lower part, as illustrated in Figure 10 U-I/1, U-V/13 and U-IX/25.

### 3.5. Deflection of the Abrasive Water Jet

The qualitative parameter deflection of the abrasive water jet *Ø* on the surface of the cut surface as a dependence *Ø* = f(*v*) is shown in Figure 11. These dependencies are presented in the form of conjugated graphs for the cases where the mass flow of the abrasive is constant (in one graph, three dependencies are plotted for three different values of pressures 300, 340 and 380 MPa).

## 4. Discussion

Specifically, the values of technological parameters *m_A_* = 170 g/min, *p* = 300 MPa and *v* = 20 mm/min (cut I/3) represent the combination for the highest values of the roughness parameters measured (*Ra* = 6.95 and *Rz* = 24.96), whereas *Ra* values range from 2.27 to 6.95 and *Rz* range from 16.02 to 24.96. The possibility to reduce the *Ra* value from 6.95 to 2.27 represents an improvement in roughness of almost 68% (approximately one third of the highest value reached).

On the other side, the values of technological parameters *m_A_* = 270 g/min, *p* = 380 MPa and *v* = 10 mm/min (cut IX/25) represent the combination for the lowest roughness parameters measured (*Ra* = 2.27 and *Rz* = 16.02). The reduction in the *Rz* value from 24.96 to 16.02 represents an improvement in roughness of almost 36%.

Mathematical models describing the functional dependence of selected technological parameters *m_A_* (x1), *p* (x2) and *v* (x3) on the quality parameters *Ra* (y) and *Rz* (y) are presented below. Figure 12 graphically indicates the significance of individual parameters. The indices of determination indicate a close approximation of the curve given by the functional dependence of the curve constructed from the evaluation of the measured values.
*Ra* y = 7.288 − 0.014 x_1_ − 0.009 x_2_ + 0.146 x_3_  R^2^_u_ = 0.733(1)
*Rz* y = 28.725 − 0.037 x_1_ − 0.018 x_2_ + 0.348 x_3_  R^2^_u_ = 0.942(2)

According to the significance graphs in Figure 12, it can be stated that the greatest influence of the monitored technological parameters on the roughness parameters *Ra* and *Rz* has almost the same share as the cutting feed rate *v* and the abrasive mass flow *m_A_*. On the other side, the smaller influence has the pump pressure *p*. By comparing the measured values of roughness with their average values *Ra* = 3.41 and *Rz* = 19.72, it can be stated that about the same roughness *Ra* and *Rz* with deviations of 0.13 can be achieved by several combinations of technological parameters, according to Table 3. By increasing the m_A_ from 220 to 270 g/min at a constant pressure of 300 MPa, it is possible to cut a third more material with almost the same roughness value *Ra* of the cut surface. By increasing the *m_A_* from 170 to 270 g/min at a constant p = 340 MPa, it is possible to cut twice the amount of material with almost the same roughness value *Rz* of the cut surface.

The values of technological parameters *m_A_* = 170 g/min, *p* = 300 MPa and *v* = 20 mm/min represent the combination for which the highest values of roughness were achieved for *Ra*_4_ = 3.36 and *Rz*_4_ = 21.69 for *Ra*_4_ ranging from 2.02 to 3.36 and *Rz*_4_ ranging from 16.19 to 21.69 (cut I/3).

On the other side, the values of technological parameters *m_A_* = 270 g/min, *p* = 380 MP and v = 10 mm/min represent the combination for which the lowest values of roughness were achieved, that is *Ra*_4_ = 2.02 and *Rz*_4_ = 16.19 from the range of *Ra*_4_ <2.02;3.36> and *Rz*_4_ <16.19;21.69> of the cut (I/3). It can also be stated that as the abrasive mass flow increases (from 170 to 220 and 270 g/min), the roughness decreases. Similarly, the roughness also decreases with increasing pump operating pressure (from 300 to 340 and 380 MPa). By reducing the feed rate (from 20 to 15 to 10 mm/min), the roughness also decreases.

Mathematical models for expressing the functional dependence of selected technological parameters *m_A_* (x1); *p* (x2) and *v* (x3) on the quality parameters *Ra*_4_ (y) and *Rz*_4_ (y) are given below. Figure 13 shows a graphical representation of the significance of individual parameters. These indices of determination indicate an approximation of the curve given by the functional dependence to the curve constructed from the measured values.
*Ra*_4_ y = 5.037 − 0.007 x_1_ − 0.003 x_2_ + 0.014 x_3_  R^2^_u_ = 0.943(3)
*Rz*_4_ y = 27.982 − 0.026 x_1_ − 0.013 x_2_ + 0.057 x_3_  R^2^_u_ = 0.918(4)

According to the significance graphs in Figure 13, it can be stated that the greatest influence on the change in roughness values *Ra*_4_ and *Rz*_4_ has a change in the feed rate of the technological head (observed at constant abrasive mass flow and pump pressure), where the interval of variations respective to lower roughness values of *Ra*_4_ range from 1.6% to 44.8% and of *Rz*_4_ from 1% to 32.8%. The second greatest effect on roughness has the value of abrasive mass flow (observed at constant pump pressure and cutting head feed rate), with an interval for the improved values ranging for *Ra*_4_ from 2.9% to 30.4% and for *Rz*_4_ from 2.3% to 19%. The pump pressure (observed at constant abrasive mass flow and cutting head feed rate) has the least effect on roughness from the selected parameters, with an interval of varying values ranging for *Ra*_4_ from 0% to 18.5% and for *Rz*_4_ from 1.1% to 14.1%.

By comparing the measured values of roughness with their average values of *Ra*_4_ = 3.15 and *Rz*_4_ = 20.95, it can be stated that about the same roughness *Ra*_4_ (with a variance of 0.07) and *Rz*_4_ (with a variance of 0.2) can be achieved by several combinations of technological parameters, according to Table 4.

The values of technological parameters *m_A_* = 170 g/min, *p* = 300 MPa and *v* = 20 mm/min represent a combination for which the worst qualitative parameter value was achieved, whereas the deflection of the abrasive water jet was observed at *Ø* = 30.1°, ranging from 9.6° to 30.1° (cut I/3). The values of technological parameters *m_A_* = 270 g/min, *p* = 380 MPa and *v* = 10 mm/min represent the combination for which the best value of the jet deflection of *Ø* = 9.6° was observed, ranging from 9.6° to 30.1° (which means an improvement of 68%) for the cut (IX/25). It can also be stated that as the cutting head feed rate decreases (from 20 to 15 and further to 10 mm/min), the jet deflection decreases. As the abrasive mass flow increases (from 170 to 220 up to 270 g/min), the jet deflection decreases. As the pump pressure increases (from 300 to 340 and 380 MPa), the jet deflection decreases.

The mathematical model for expressing the functional dependence of the selected technological parameters *m_A_* (x_1_), *p* (x_2_) and *v* (x_3_) on the quality parameter represented by deflection of the abrasive water jet *Ø* (y) is given below. Figure 14 provides a graphical representation of the significance of individual parameters. These indices of determination indicate an approximation of the curve given by the functional dependence to the curve constructed from the measured values.
*Ø* y = 38.118 − 0.071 x_1_ − 0.048 x_2_ + 0.702 x_3_  R^2^_u_ = 0.873(5)

Based on the differences of the specific measured values of the jet deflection at the mutual concurrent influence of the monitored technological parameters (*m_A_*, *p* and *v*), and according to the graph of significance in Figure 14, it can be stated that the greatest influence on the variations of jet deflection values is the change of the cutting head feed rate (observed at constant *m_A_* and *p*), where the interval of changes towards the lower jet deflection values was from 8.2% to 42.5%. The second largest, which had almost the same effect on the jet deflection as the feed rate of the cutting head, has a change of the abrasive mass flow (observed at constant *p* and *v*) with the change’s interval towards better values, from 6.9% to 42.7%. The pump pressure (observed at constant *m_A_* and *v*) has a smaller effect on the jet deflection with a variation interval of 2.4% to 30.7%.

When comparing the measured values of jet deflections with their average value of 16.8°, it can be stated that almost the same jet deflection with a scattering of only 1.9° can be achieved by several combinations of technological parameters *m_Amin_–p_min_–v_min_*~*m_Amin_–p_max_–v_med_*~*m_Amed_–p_min_–v_med_*_~_*m_Amed_–p_med_–v_med_*~*m_Amax_–p_max_–v_max_* (*med* denotes medium value). Table 5 summarizes this more clearly with specific parameter values and measured jet deflection values.

### 4.1. Recommendations for Values of Roughness Ra, Rz and Deflection of Abrasive Water Jet Ø

The recommended values of technological parameters for achieving the minimum and maximum experimentally measured values of roughness *Ra*, *Rz* and *Ø* are clearly summarized in Table 6. The significance of the monitored technological parameters influencing the quality of the cut surfaces is given in Table 7.

### 4.2. Recommendations for Roughness Values Ra_4_ and Rz_4_

For 40 mm thick sheets, in order to achieve low roughness (*Ra*_4_ about 2 and *Rz*_4_ about 16) on the cut surface at a distance of 4 mm from the upper cutting edge (Figure 3), and to achieve a small deflection of the abrasive water jet (*Ø* below 10°), it is recommended to use values of technological parameters *m_A_* = 270 g/min, *p* = 380 MPa and *v* = 10 mm/min. High roughness values (*Ra*_4_ about 3.4 and *Rz*_4_ about 21.7) at a distance of 4 mm from the upper cutting edges is achieved at technological parameters *m_A_* = 170 g/min, *p* = 300 MPa and *v* = 20 mm/min. The most influential (most important) parameter for the change of roughness is the feed rate of the cutting head, followed by the mass flow of the abrasive, and the least influential parameter is the pump pressure.

## 5. Conclusions

The article is aimed at the study of the influence of three selected technological parameters (abrasive mass flow, pump pressure and cutting/technological head feed rate on selected quality parameters of cut surfaces (surface roughness *Ra*, *Rz*, *Ra*_4_, *Rz*_4_ and jet deflection *Ø*)) after cutting 40 mm thick Hardox 500 steel sheet samples using the AWJ technological system. The presented results apply to the specific conditions of the experiments as described in the article. The obtained results of measurements and analyses allow to formulate the following conclusions:−The most significant improvements regarding the roughness of the cut surfaces of 40 mm thick Hardox sheets are achieved by reducing the feed rate of the cutting head and increasing the mass flow of the abrasive. The same applies to the deflection of the abrasive water jet. Pump pressure parameter is less effective to change the roughness as well as the deflection of the abrasive water jet;−Lower roughness values of *Ra*, *Rz*, *Ra*_4_ and *Rz*_4_ in the samples were caused by lower cutting feed rate, but at the same time, the highest used value of the feed rate *v* = 20 mm/min can be considered as the limit if it acts in combination with the lowest values of the other two technological parameters mA and p. If this limit value is exceeded, it may not be possible to perform a complete cut of the sheet;−The feed rate of the cutting head has the greatest influence on the deflection of the jet because half the rate can compensate for the simultaneous change of mass flow and pressure from the maximum (Table 5, combination of parameters No. 5) to the level of their medium value (No. 4), as well as reduction in the feed rate from maximum (No. 3) to mean value (No. 5) compensates for the reduction in pressure from maximum to minimum and at the same time the mass flow from maximum to medium value (No. 3), as well as only the change of mass abrasive flow from maximum to minimum (No. 2). Similarly, reducing the feed rate from maximum to minimum compensates for the change in mass flow and pressure maxima (No. 5) to their minimums (No. 1). The pump pressure parameter has the least effect on the jet deflection;−The surface quality of the upper zone of the cutting material is better than the quality of the lower zone.−The evaluation of experiments presented in this article does not elaborate all related issues of the problems of examining the influence of selected technological parameters in terms of surface quality. It is therefore necessary to continue in this important research, especially in the context of AWJ process optimization for industrial applications. The directions of future research within the topic may comprise, for example, expanding the set of experiments for other sheet thicknesses of Hardox 500 steel or other types of Hardox steel (400, 450, 550, 600, Extreme, HiTuf or other hard-to-machine material). An alternative way is to investigate the dependence of other types of abrasive on the quality parameters of the cut surface. Finally, it could also be enwidening to research the influence of selected technological (*m_A_*, *p*, *v*,…) and quality parameters (*Ra*, *Rz*, *Ø*,…) at wider intervals of their values.

## Figures and Tables

**Figure 1 materials-15-01381-f001:**
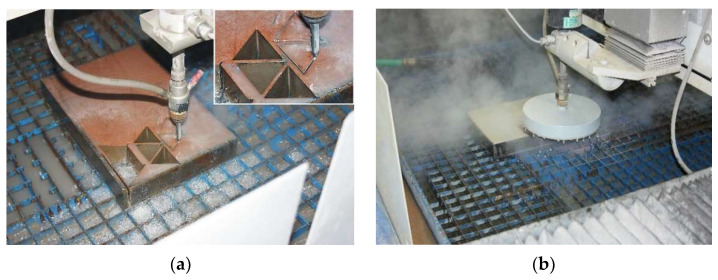
(**a**) Cutting 40 mm thick Hardox 500 sheet metal; (**b**) usage of protective cylinder.

**Figure 2 materials-15-01381-f002:**
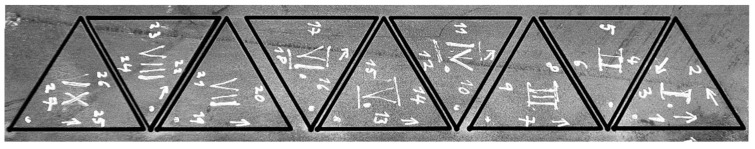
Example of sample marking.

**Figure 3 materials-15-01381-f003:**
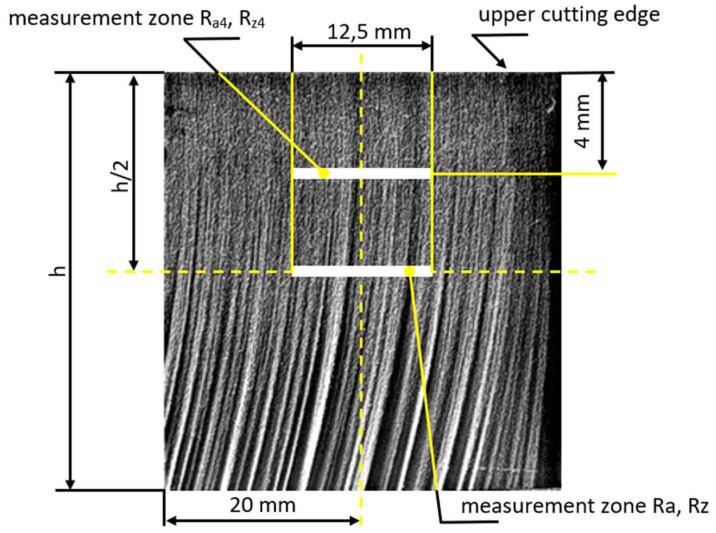
Roughness measuring zones for the parameters *Ra*, *Rz*, *Ra*_4_, and *Rz*_4_ on the cut surfaces of the samples.

**Figure 4 materials-15-01381-f004:**
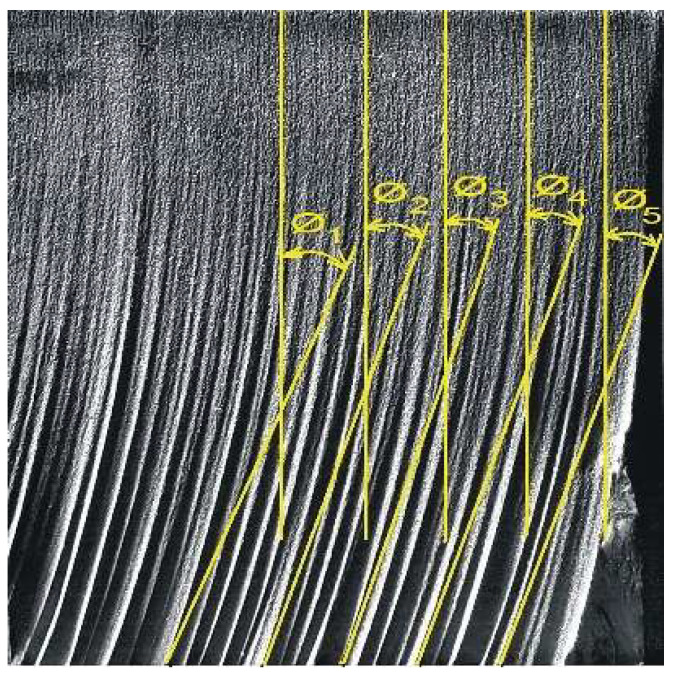
Localization of the measurement of the deflection of the abrasive water jet trace on the cut surface.

**Figure 5 materials-15-01381-f005:**
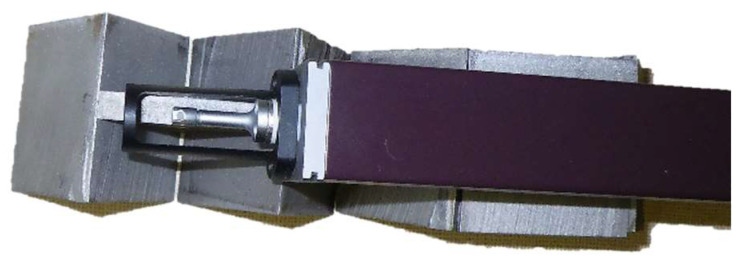
Surftest SJ-301 device measuring the roughness of the cut surface.

**Figure 6 materials-15-01381-f006:**
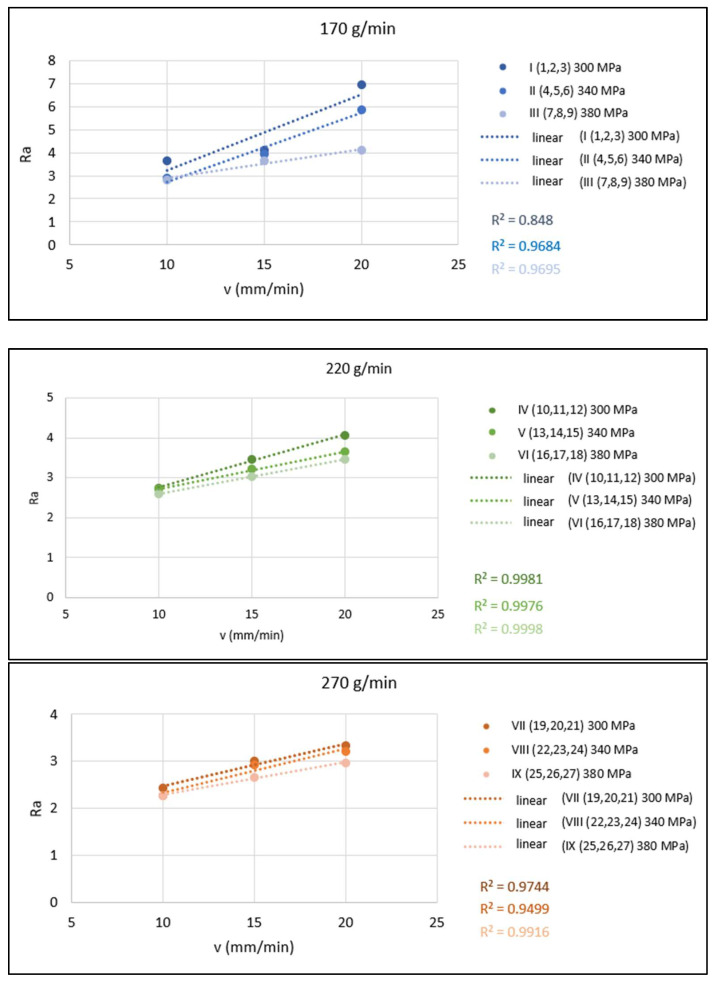
Graphical evaluation of 1-parametric dependences of *Ra* = f(*v*) for *m_A_* = 170, 220 and 270 g/min.

**Figure 7 materials-15-01381-f007:**
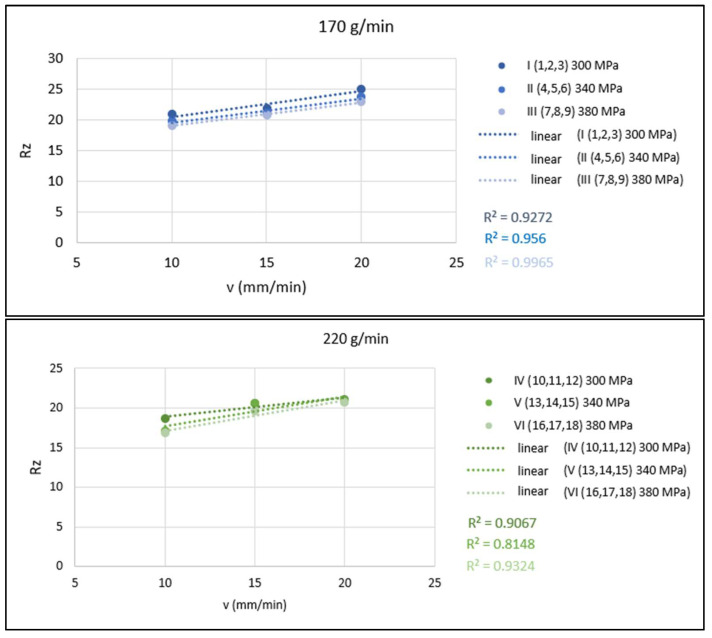
Graphical evaluation of 1-parametric dependencies of *Rz* = f(*v*) for *m_A_* = 170, 220 and 270 g/min.

**Figure 8 materials-15-01381-f008:**
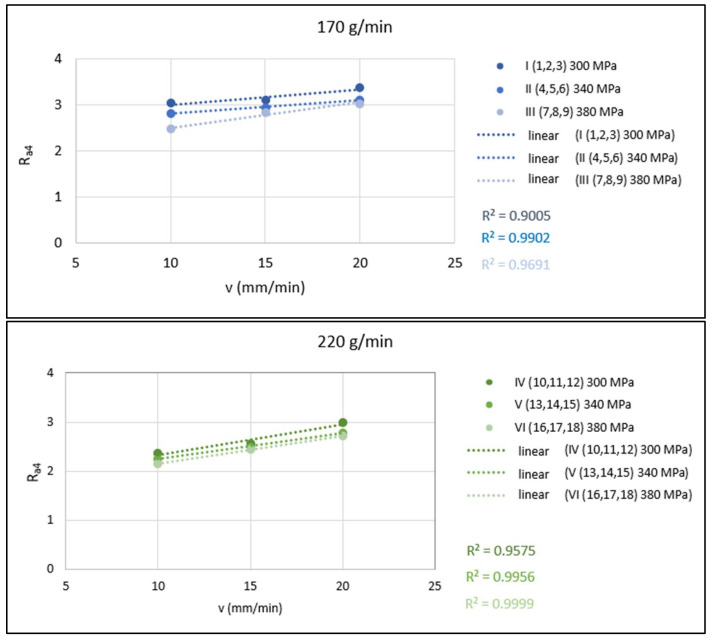
Graphical evaluation of 1-parametric dependencies of *Ra*_4_ = f(*v*) for *m_A_* = 170, 220 and 270 g/min.

**Figure 9 materials-15-01381-f009:**
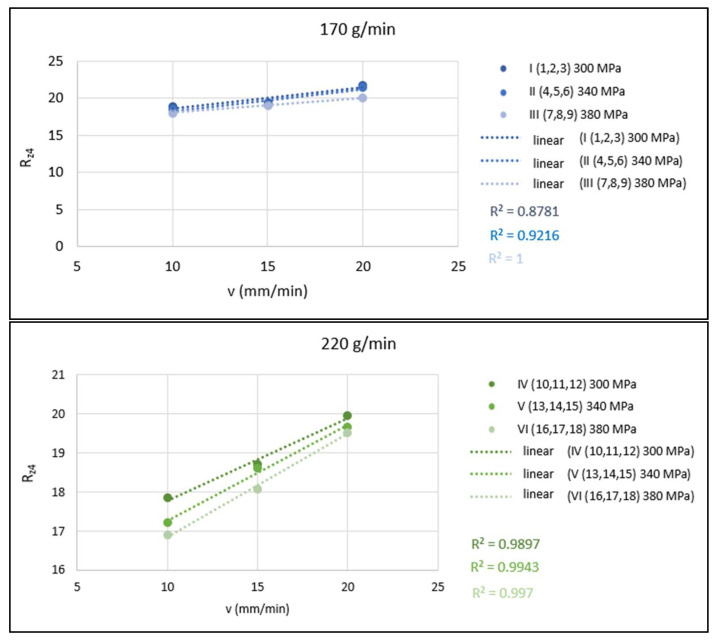
Graphical evaluation of 1-parametric dependencies of *Rz*_4_ = f(*v*) for *m_A_* = 170, 220 and 270 g/min.

**Figure 10 materials-15-01381-f010:**
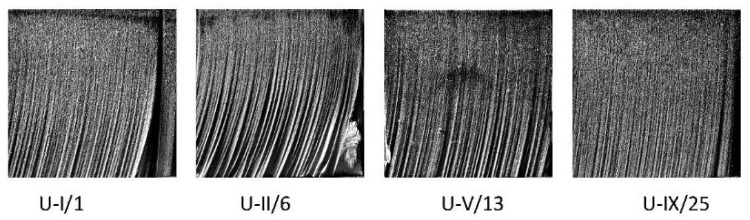
Photographic images of cut surfaces.

**Figure 11 materials-15-01381-f011:**
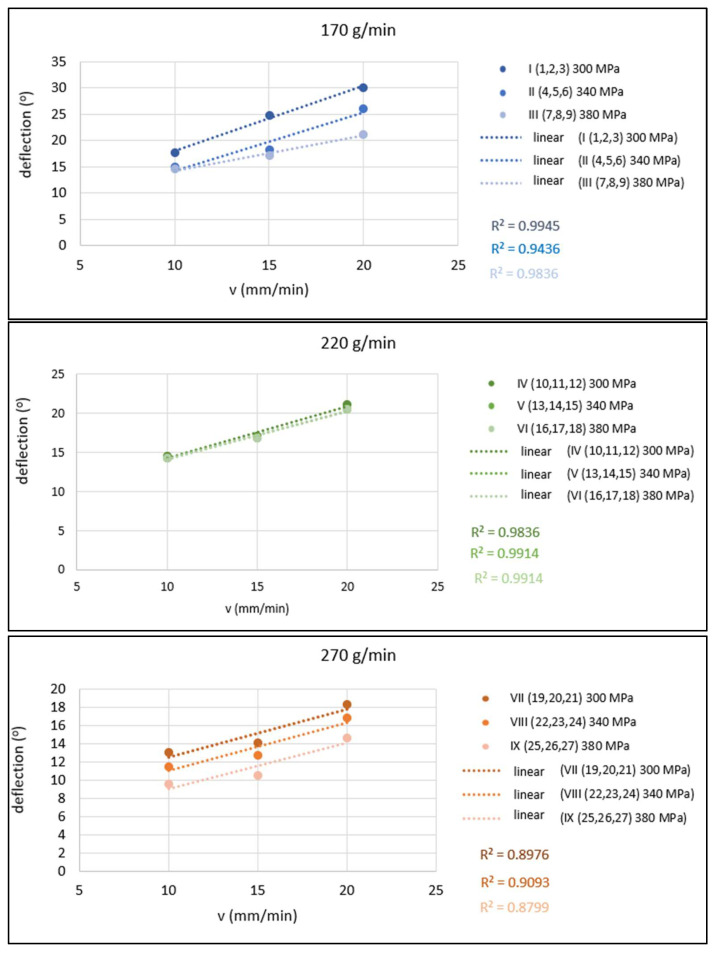
Graphical evaluation of 1-parametric dependencies of *Ø* = f(*v*) for *m_A_* = 170, 220 and 270 g/min.

**Figure 12 materials-15-01381-f012:**
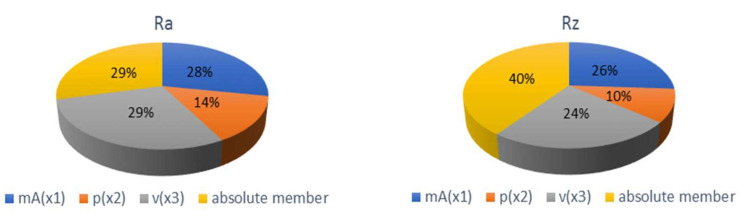
Graph of significance of mathematical model parameters for *Ra* and *Rz* roughness parameters.

**Figure 13 materials-15-01381-f013:**
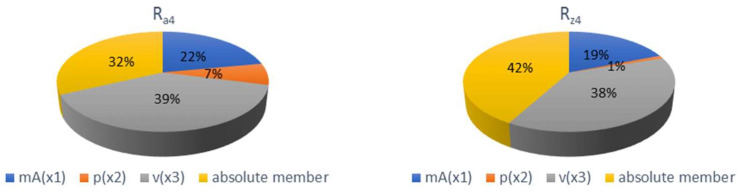
Graph of significance of mathematical model parameters for *Ra*_4_ and *Rz*_4_ roughness parameters.

**Figure 14 materials-15-01381-f014:**
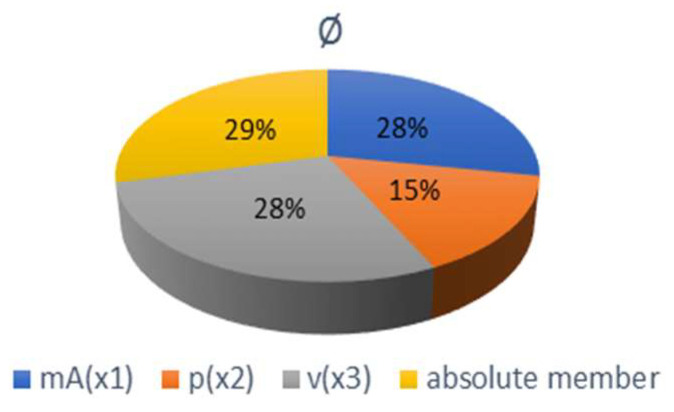
Graph of significance of mathematical model parameters for deflection *Ø*.

**Table 1 materials-15-01381-t001:** Values of the deflection of the abrasive water jet track *Ø* and the roughness parameters *Ra* and *Rz* as measured on the cut surfaces of the samples.

Sample Identification		Technological Parameters			Qualitative Parameters							
Sample No.	Sample Cut No.	*m_A_* [g/min]	*p* [MPa]	*v* [mm/min]	Repeated Measurements for Evaluation *Ø* [°]					*Ø* [°]	*Ra*	*Rz*
					*Ø* _1_	*Ø* _2_	*Ø* _3_	*Ø* _4_	*Ø* _5_			
I	1	170	300	10	17.4	18.1	17.5	18.0	17.5	17.7	3.65	20.84
	2	170	300	15	20.8	28.0	25.2	25.1	24.2	24.7	4.09	21.90
	3	170	300	20	34.9	26.8	28.5	28.0	32.1	30.1	6.95	24.96
II	4	170	340	10	15.0	14.2	15.8	14.2	15.9	15.0	2.90	19.79
	5	170	340	15	17.5	19.0	18.2	17.5	18.6	18.2	3.92	21.00
	6	170	340	20	22.8	25.4	30.0	29.5	23.0	26.1	5.87	23.64
III	7	170	380	10	14.0	14.8	14.6	14.8	14.4	14.5	2.83	19.11
	8	170	380	15	17.4	16.8	17.0	17.0	17.3	17.1	3.66	20.81
	9	170	380	20	22.1	19.7	21.7	19.4	22.9	21.2	4.10	22.90
IV	10	220	300	10	14.1	14.2	14.5	14.3	14.3	14.3	2.75	18.67
	11	220	300	15	16.5	17.2	17.1	16.8	17.1	16.9	3.46	20.56
	12	220	300	20	21.7	19.8	20.3	19.6	21.3	20.5	4.07	21.10
V	13	220	340	10	14.3	13.6	13.9	13.7	13.8	13.9	2.71	17.14
	14	220	340	15	15.4	16.0	16.1	15.8	15.5	15.8	3.22	20.60
	15	220	340	20	19.0	21.1	19.3	18.8	21.0	19.8	3.65	20.93
VI	16	220	380	10	14.0	13.4	13.1	14.1	13.0	13.5	2.60	16.83
	17	220	380	15	15.1	15.7	15.3	15.4	15.4	15.4	3.02	19.66
	18	220	380	20	18.1	19.6	18.6	18.8	18.7	18.8	3.46	20.69
VII	19	270	300	10	13.3	13.1	12.5	12.2	13.7	13.0	2.44	16.52
	20	270	300	15	15.0	13.5	13.8	14.7	13.6	14.1	3.01	19.30
	21	270	300	20	17.6	18.2	19.1	19.0	17.8	18.3	3.33	20.10
VIII	22	270	340	10	11.7	10.7	12.1	12.2	11.0	11.5	2.28	16.25
	23	270	340	15	12.9	12.5	12.4	12.9	12.8	12.7	2.93	18.29
	24	270	340	20	17.1	16.7	16.5	16.8	16.8	16.8	3.21	19.77
IX	25	270	380	10	10.2	9.4	9.3	9.1	9.8	9.6	2.27	16.02
	26	270	380	15	10.1	10.4	11.1	10.4	10.5	10.5	2.67	16.77
	27	270	380	20	15.2	14.7	14.1	13.9	14.9	14.6	2.96	18.40
**Average values of *Ø, Ra* and *Rz*, respectively**										**16.8**	**3.41**	**19.72**

**Table 2 materials-15-01381-t002:** Values of the roughness parameters *Ra*_4_ and *Rz*_4_ as measured on the cut surfaces of the samples at a distance of 4 mm.

Sample Identification		Technological Parameters			Qualitative Parameters	
Sample No.	Sample Cut No.	*m_A_* [g/min]	*p* [MPa]	*v* [mm/min]	*Ra* _4_	*Rz* _4_
I	1	170	300	10	3.03	18.87
	2	170	300	15	3.10	19.37
	3	170	300	20	3.36	21.69
II	4	170	340	10	2.81	18.44
	5	170	340	15	2.93	19.17
	6	170	340	20	3.10	21.39
III	7	170	380	10	2.47	18.01
	8	170	380	15	2.83	19.01
	9	170	380	20	3.02	19.99
IV	10	220	300	10	2.36	17.86
	11	220	300	15	2.56	18.72
	12	220	300	20	2.99	19.95
V	13	220	340	10	2.26	17.23
	14	220	340	15	2.49	18.61
	15	220	340	20	2.78	19.67
VI	16	220	380	10	2.16	16.90
	17	220	380	15	2.44	18.09
	18	220	380	20	2.71	19.53
VII	19	270	300	10	2.11	16.81
	20	270	300	15	2.29	17.92
	21	270	300	20	2.68	18.96
VIII	22	270	340	10	2.07	16.64
	23	270	340	15	2.24	17.54
	24	270	340	20	2.53	18.92
IX	25	270	380	10	2.02	16.19
	26	270	380	15	2.22	17.23
	27	270	380	20	2.50	18.78
**Average values of *R_a_*_4_ and *R_z_*_4_**					**3.15**	**20.95**

**Table 3 materials-15-01381-t003:** Example of combinations of values of technological parameters for achieving approximately equal (average) roughness *Ra* = 3.41; *Rz* = 19.72.

Combination of Parameters	Technological Parameter	Value	*Ra*	Cutting Surface Identification	*Rz*	Cutting Identification
1	*m_A_*	220 g/min	3.46	IV/11	19.79	II/4
	*p*	300 MPa				
	*v*	15 mm/min				
2	*m_A_*	220 g/min	3.46	VI/18	19.66	VI/17
	*p*	380 MPa				
	*v*	20 mm/min				
3	*m_A_*	270 g/min	3.33	VII/21	19.77	VIII/24
	*p*	300 MPa				
	*v*	20 mm/min				

**Table 4 materials-15-01381-t004:** Example of combinations of values of technological parameters for achieving approximately equal (average) roughness *Ra*_4_ = 3.15; *Rz*_4_ = 20.95.

Combination of Parameters	Technological Parameter	Value	*Ra* _4_	Cutting Surface Identification	Technological Parameter	Value	*Rz* _4_	Cutting Identification
1	*m_A_*	170 g/min	3.10	I/2	*m_A_*	170 g/min	19.99	III/9
	*p*	300 MPa			*p*	380 MPa		
	*v*	15 mm/min			*v*	20 mm/min		
2	*m_A_*	170 g/min	3.10	II/6	*m_A_*	170 g/min	21.39	II/6
	*p*	340 MPa			*p*	340 MPa		
	*v*	20 mm/min			*v*	20 mm/min		
3	*m_A_*	170 g/min	3.03	I/1	*m_A_*	220 g/min	19.95	IV/12
	*p*	300 MPa			*p*	300 MPa		
	*v*	10 mm/min			*v*	20 mm/min		

**Table 5 materials-15-01381-t005:** Combinations of technological parameter values selected from the evaluated ranges that have a similar effect to achieving approximately the same deflection of the abrasive water jet *Ø* (16.8°).

Combination of Parameters	Min	Medium	Max	Technological Parameter Value	Jet Deflection	Cutting Identification
1	*m_Amin_*			170 g/min	17.7°	I/1
	*p_min_*			300 MPa		
	*v_min_*			10 mm/min		
2	*m_Amin_*			170 g/min	17.1°	III/8
			*p_max_*	380 MPa		
		*v_med_*		15 mm/min		
3		*m_Amed_*		220 g/min	16.9°	IV/11
	*p_min_*			300 MPa		
		*v_med_*		15 mm/min		
4		*m_Amed_*		220 g/min	15.8°	V/14
		*p_med_*		340 MPa		
		*v_med_*		15 mm/min		
5			*m_Amax_*	270 g/min	16.8°	VIII/24
			*p_max_*	380 MPa		
			*v_max_*	20 mm/min		

**Table 6 materials-15-01381-t006:** Recommended values of technological parameters for achieving minimum and maximum experimentally measured values of parameters *Ra*, *Rz* and *Ø*.

	Technological Parameter			Qualitative Parameter			
	*m_A_*	*p*	*v*	*Ra*	*Rz*	*Ø*	Quality (Surface Roughness), Application
for minimum measured values	270 g/min	380 MPa	10 mm/min	2.27	16.02	9.6°	low roughness, without/easy further processing
for maximum measured values	170 g/min	300 MPa	20 mm/min	6.95	24.96	30.1°	very high roughness, material cutting

**Table 7 materials-15-01381-t007:** Significance of technological parameters influencing the quality of the cut surface.

Technological Parameter	Significance of the Technological Parameter	Technological Conditions		
		*m_A_* [g/min]	*p * [MPa]	*v* [mm/min]
*m_A_*	High	170; 220; 270	300; 340; 380	10; 15; 20
*p*	Low			
*v*	High			

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
