# Peer review of "Assessment of the Influence of Selected Technological Parameters on the Morphology Parameters of the Cutting Surfaces of the Hardox 500 Material Cut by Abrasive Water Jet Technology"

_materials, 2022, doi:10.3390/ma15041381_

Round 1

Reviewer 1 Report

Very interesting research and experimental work in the field of production engineering and application of abrasive jet processing.

Perhaps in the introduction it could be emphasized that the application of AWJ material processing in relation to other processes, and that is that there is no thermal change of material in the cutting zone, which is present in some other processes.

I would also have the following questions for authors you would like to include in the text of your paper:

  1. Why was the Australian grenade used in this experiment?
  2. What is the accuracy of dimensional measurements?
  3. Why did you decide to research Hardox and 40mm thickness?
  4. Why was the roughness measured at a distance of 4 mm from the top cutting edge?

While reading the paper, I noticed some technical errors:

  1. Row 178: instead of pictures 4 and 5 should be 3 and 4.
  2. Row 188/189: To be added at the end of the table title: at a distance of 4 mm.
  3. Line 323: It is necessary to define which image. Now stands Fig. xxx.

Give comments on the values of the coefficient of determination (R2) whose values are shown in Figures 6,7,8,9 and 11.

Author Response

Dear Reviewer,

we are very grateful for your questions and remarks aimed at improving our article.

We tried to respond to them as best we could and added recommended improvements and explanations into the manuscript. For further details, please see the attachment.

Best regards,
Authors

Reviewer 2 Report

Manuscript is very interesting, fits the journal scope and can be considered for publishing after clarifying the following suggestions:

  • why the Ra4 roughness was measured at a depth of 4mm from the edge and not, for example, at 5mm?
  • a new set of nozzles was used for the tests? there is no information about it anywhere 
  • I have reservations about the presentation of the test results, e.g. Figure 6-11 Are two-dimensional graphs appropriate? Perhaps a better solution would be to present the results in three-dimensional charts. 

Author Response

Dear Reviewer,

we are very grateful for your questions and comments aimed at improving the article.

We tried to respond to them as best we could and added recommended improvements and explanations into the manuscript. For further details, please see the attachment.

Best regards,                                                                                                     Authors
